# Transcriptional Stochasticity as a Key Aspect of HIV-1 Latency

**DOI:** 10.3390/v15091969

**Published:** 2023-09-21

**Authors:** Alexia Damour, Vera Slaninova, Ovidiu Radulescu, Edouard Bertrand, Eugenia Basyuk

**Affiliations:** 1MFP UMR 5234 CNRS, Université de Bordeaux, 33076 Bordeaux, France; alexia.damour@u-bordeaux.fr; 2IGH UMR 9002 CNRS, Université de Montpellier, 34094 Montpellier, France; vera.slaninova@cnrs.fr; 3LPHI, UMR 5294 CNRS, University of Montpellier, 34095 Montpellier, France; ovidiu.radulescu@umontpellier.fr

**Keywords:** HIV-1 latency, transcription, transcriptional noise, transcriptional burst, promoter, transcription factor, chromatin, integration site, modeling

## Abstract

This review summarizes current advances in the role of transcriptional stochasticity in HIV-1 latency, which were possible in a large part due to the development of single-cell approaches. HIV-1 transcription proceeds in bursts of RNA production, which stem from the stochastic switching of the viral promoter between ON and OFF states. This switching is caused by random binding dynamics of transcription factors and nucleosomes to the viral promoter and occurs at several time scales from minutes to hours. Transcriptional bursts are mainly controlled by the core transcription factors TBP, SP1 and NF-κb, the chromatin status of the viral promoter and RNA polymerase II pausing. In particular, spontaneous variability in the promoter chromatin creates heterogeneity in the response to activators such as TNF-α, which is then amplified by the Tat feedback loop to generate high and low viral transcriptional states. This phenomenon is likely at the basis of the partial and stochastic response of latent T cells from HIV-1 patients to latency-reversing agents, which is a barrier for the development of shock-and-kill strategies of viral eradication. A detailed understanding of the transcriptional stochasticity of HIV-1 and the possibility to precisely model this phenomenon will be important assets to develop more effective therapeutic strategies.

HIV-1 infections can be effectively controlled using combined antiretroviral therapy (ART). Yet, viral latency is a major challenge for viral eradication from patients. Latent viruses harbored by cell and tissue reservoirs are invisible to the immune system but inevitably cause rebound upon ART interruption. The establishment and reactivation of latency have stochastic components, and HIV-1 has evolved mechanisms to benefit from the stochastic nature of transcription. In this review, we will present the current knowledge on HIV-1 transcriptional noise and its role in latency regulation.

## 1. Transcriptional Noise Is a Ubiquitous Phenomenon

Cells display phenotypic differences even within isogenic populations occupying the same environment. These differences are often caused by stochastic variations in transcription, also called transcriptional noise, which lead to fluctuations in mRNA and protein levels [1,2,3]. Noise is described as the distribution of the RNA or protein number per cell in the population and its intensity can be measured using metrics of variability such as the coefficient of variation (the standard deviation from the mean ratio) [4,5] or the Fano factor (the variance from the mean ratio) [6]. 

Sources of noise can be extrinsic or intrinsic [3,7]. Extrinsic noise comes from cell-wide perturbations such as changes in cell size, the cell cycle phase and signaling pathways, or in the global concentration of generic factors involved in transcription, RNA processing, translation and degradation [3,8]. Extrinsic noise varies from cell to cell but affects all the genes in one cell equally, therefore causing correlated expressions of the two gene copies in diploid cells. In contrast, intrinsic noise arises from the inherent stochasticity of the biochemical reactions leading to gene expression, as these involve a small number of molecules such as a single promoter DNA and a few transcription factor molecules. As such, intrinsic noise affects all the gene copies within one cell differently, causing their uncorrelated expression. Example of processes generating intrinsic noise include for instance the random binding and dissociation of transcription factors to a promoter or an enhancer. 

In the simplest model, when a promoter is at steady state and constitutively active, RNA polymerases can bind to it and stochastically initiate transcription, leading to a Poissonian distribution of the RNA number per cell [9]. Such a simple model does not correspond to experimental measurements in many cases and a more complex model has been proposed, in which the promoter stochastically switches between an active (ON) and an inactive (OFF) state instead of being active all the time [2,10]. This model, also called the random telegraph model, describes transcriptional bursting, which corresponds to episodes of RNA production during the ON state of the promoter, alternating with the absence of transcription during the OFF state (for a review, see [11]). The distribution of the RNA number per cell in this case is more complex than Poissonian and can be approximated using the beta or gamma distributions [12,13]. Bursting was shown to be a predominant mode of gene expression in all organisms from bacteria to mammals [14,15,16,17,18]. In mammals, however, the random telegraph model with a simple ON/OFF switch is often not able to recapitulate the transcriptional fluctuations observed via direct RNA imaging, and additional promoter states are required to fit the experimental data [19,20,21,22,23].

Gene expression noise plays an important role in the phenotypic diversity of cells and organisms. It has been shown to have important consequences for biological processes as diverse as the adaptation of bacterial and yeast populations to environmental changes, cell fate specification during organism development, responses to signaling and to pathogens, as well as the penetrance of mutations and the phenotypic diversity of twins [24,25,26,27,28,29]. Transcriptional noise can be harmful as it can compromise crucial biological processes such as cell differentiation, development and response to stimuli, which should be robust and reproducible. Therefore, cells have evolved various mechanisms for transcriptional noise buffering, such as the stabilization of gene expression during development through transcriptional [30] or post-transcriptional regulation [31,32], the control of signaling responses by parallel feedback loops [33] or the optimization of transcriptional responses to stimuli [34]. On the other hand, transcriptional noise can be beneficial, for instance as a “bet-hedging” strategy in a population of cells, helping cells to adapt in a changing environment or to determine a particular fate [1,35,36]. HIV-1 benefits from transcriptional noise by regulating the decision between acute or latent infection, the balance of which enables the virus to both persist in the infected organism and to be transmitted [37]. More generally, transcriptional noise impacts different pathogens, including the switch between lysogeny and lysis of the bacteriophage lambda, and the latency regulation of HSV, SV40 and CMV viruses [38,39,40,41]. In these cases, transcriptional noise and virally driven autoregulatory feedback loops, positive or negative, play key roles in establishing long-term infection. 

## 2. Key Features of HIV-1 Latency 

During latency, intact replication-competent HIV-1 proviruses are integrated in the genome of the host cell but do not produce infectious viral particles. These latent viruses can, however, be reactivated, causing viral rebound [42]. Upon ART interruption in patients, reactivation of latent proviruses leads to a strong increase in viral loads within two to eight weeks of treatment arrest [43,44,45], preventing viral eradication. In the infected organism, silent proviruses are harbored by a set of cells that together form the latent reservoir (for a review, see [46,47]). This reservoir comprises different cell types and anatomical sites where the replication-competent viruses accumulate and are stably maintained [47,48]. The main reservoir of latent HIV-1 is composed of resting memory CD4^+^ T cells and is formed within a few days after infection [49,50,51,52,53]. These cells are long-lived, yielding an estimate of 3.7 years for the half-life of the latent reservoir, and an eradication time of 60 years, explaining why patients should receive life-long ART [50]. Other cellular reservoirs of latent HIV-1 are mononuclear macrophages, dendritic cells, hematopoietic progenitor cells, astrocytes and epithelial cells [46]. The viral reservoir is maintained predominantly through clonal proliferation of latent CD4^+^ T cells [54,55,56], although replenishment through a low-level of ongoing viral replication under ART was also reported [57]. While the latter mode of reservoir maintenance is still debated [58,59], the viral blips in patients seem to be associated with a large HIV-1 reservoir and its slow decay [60].

HIV-1 latency is mainly transcriptional, with few viral RNAs produced in latent cells [42]. Latency exit involves an initial activation of transcription, which is then amplified by a positive feedback loop mediated by the viral transactivator of transcription, Tat. Different triggers have been proposed to drive latency exit. A prevalent one is the cellular activation of memory T cells by external cues such as antigens, which act as an extrinsic trigger to stimulate HIV-1 transcription (for a review, see [61]). A more recent theory suggests that intrinsic transcriptional noise might lead to viral expression bursts that could be sufficiently strong to turn on the Tat-positive feedback loop, causing a sustained activation of transcription [62,63,64,65,66]. However, little is currently known about how viral transcription occurs in latent cells from HIV-1 patients, and it is unclear how frequent such spontaneous bursts could be. Nevertheless, it has been clearly demonstrated that latency exit is stochastic in T cells from patients. In a seminal paper, the Siliciano lab showed that only a fraction of replication-competent proviruses in CD4^+^ T lymphocytes from HIV-1 patients are reactivated under complete T cell activation with PHA (PhytoHemAgglutinin), and that nonactivated proviruses can be reactivated in repeated rounds of PHA activation [67]. This finding was confirmed by follow-up studies [68,69], and it was in line with previous findings that observed a stochastic activation of HIV-1 in cell lines [63,70]. Based on these observations, it is now widely accepted that stochastic reactivation poses a major challenge to shock-and-kill strategies of viral eradication. Therefore, even if the activation of quiescent memory T cells is a key event triggering latency exit, the stochasticity of viral expression plays an important role in this process as detailed below. 

## 3. The HIV-1 Promoter and the Tat-Positive Feedback Loop

HIV-1 is silent in latent cells and its transcription can be activated by a multitude of cellular factors and by the viral protein Tat. Tat strongly activates the viral promoter, thereby creating a positive feedback loop that is essential for its full activation (for a review, see [71,72]). The presence or absence of Tat creates a switch that controls the latent and acute states of HIV-1: acutely infected cells express Tat at a high level, while it is absent or present in minute amounts in latent cells.

The HIV-1 LTR harbors a compact promoter that is controlled at multiple levels, with two key regulatory processes being chromatin remodeling and polymerase pausing (Figure 1). In basal conditions when the viral protein Tat is absent, RNA polymerase II (RNAPII) can initiate transcription at the viral promoter but it transcribes only about 60–80 nucleotides before entering into a paused state. The Negative Elongation Factor (NELF) and the DRB Sensitivity-Inducing Factor (DSIF) are required for RNAPII pausing (for a review, see [72]). In addition, the presence of the +1 nucleosome (Nuc-1), positioned downstream of the transcription start site (TSS), also participates in the blocking of RNAPII elongation [73,74]. As a consequence, transcription results in the production of short transcripts with only a very low amount of full-length RNAs, which are nevertheless spliced, exported and used to produce Tat [71,72]. The first 60 nucleotides of the HIV-1 RNAs fold into a stable stem loop structure called TAR (Transactivation Response) element (Figure 1, middle panel) [75,76]. Tat binds to TAR and recruits the cellular Positive Transcription Elongation Factor b (P-TEFb), composed of the kinase CDK9 and Cyclin T1 (CycT1) [77]. Tat also interacts with CycT1, which directly binds to TAR, promoting the formation of a high-affinity ternary complex [78]. P-TEFb often comes as part of the super-elongation complex (SEC), containing the AFF4 and ELL proteins that further stimulate TAR binding and P-TEFb kinase activity [79,80]. P-TEFb phosphorylates NELF, DSIF and the C-terminal domain of RNAPII, thus disrupting the pausing complex and promoting productive elongation (Figure 1, bottom panel) [81,82,83,84].

The second major activity of Tat is remodeling the nucleosomal structure of the HIV-1 promoter [85,86]. In the absence of Tat, the viral promoter contains Nuc-0, located ~450 bases upstream of the TSS, and Nuc-1, positioned right after the TSS and associated with transcriptional repression [73,87] (Figure 1). During activation of viral transcription, these nucleosomes become acetylated and Nuc-1 is displaced, facilitating the passage of RNAPII. Tat plays a key role in this event through the recruitment of a number of chromatin-remodeling factors, such as the Histone Acetyl-Transferases (HATs) p300/CBP and PCAF, and the SWI/SNF remodeling complexes BAF and P-BAF [87,88,89,90]. Interestingly, two links between the nucleosome occupancy at promoters and RNAPII pausing have been suggested. First, the presence of a paused RNAPII has been proposed to prevent the nucleosomal occlusion of promoters, possibly by competing for promoter occupancy [91]. Second, pausing near the promoter and at the downstream +1 nucleosome now appear to be two distinct steps during the transition to productive elongation (for a review, see [92]). Possibly, promoter proximal pausing may orchestrate the recruitment of the factors necessary for productive elongation [93] and for progressing through the downstream nucleosomes [94]. The dual role of Tat in pause release and chromatin remodeling likely explains the high transcriptional activity of HIV-1 in activated conditions. 

Between Nuc-0 and Nuc-1, there is a DNAse-hypersensitive region called DHS I. This region contains binding sites for a number of transcription factors including Sp1, NF-κB, NFAT and AP-1, as well as a TATA box and a highly active initiator sequence. The HIV-1 core promoter has three Sp1 and between one and four NF-κB binding sites, depending on the viral subtype (Figure 1). Sp1 and TBP play a crucial role in the basal level of HIV-1 transcription *in vivo*, while the other factors regulate viral transcriptional activity and its response to cellular stimuli (reviewed in [71]). 

## 4. Tat Feedback Induces Stochastic Switches between High and Low Viral Expression States

In a pioneering study, Weinberger and collaborators showed that the stochasticity of the Tat feedback loop controls the determination of active and latent states [63]. They developed an HIV-1 reporter expressing GFP together with an IRES-driven Tat (minivirus LTR-GFP-IRES-Tat-LTR, or LGIT; Figure 2A). Jurkat cells were transduced with this reporter and cells with a low level of GFP expression were sorted via FACS to obtain individual clones with a single integration of the reporter. After 3 weeks in culture, the expanded clones displayed three different phenotypes: 75% of clones showed no GFP expression, 2% had high GFP expression and the remaining 23% consisted of clones with a variegated phenotype containing two cell populations—high- and low-GFP-expressing cells (Figure 2B,C). The authors coined the term phenotypic bifurcation (PheB) to describe this variegated phenotype with bimodal GFP expression [63]. It was further found that subcloning of high- or low-expressing cells from such a clone regenerated populations with bimodal expression, indicating that cells spontaneously switched between high- and low-GFP-expression states [70,95,96] (Figure 2D). Importantly, an HIV-1 reporter without IRES-Tat expressed only a low level of GFP without PheB [63], while cells expressing Tat in trans were all GFP-positive [70]. These results suggested that expression from the HIV-1 promoter was stochastic and that the Tat feedback loop amplified this noise to create metastable states with high and low GFP (and Tat) expression. These metastable states lasted for days and were proposed to be analogous to acute and latent states in the cells of patients [63,70]. The existence of a basal noise from the HIV-1 promoter was suggested by an earlier study [97] and further supported by time-lapse observations of single cells infected with LTR-GFP vectors without the Tat feedback loop, which displayed fluctuations in GFP level around the mean [16,98,99,100]. Overall, these studies demonstrated the key importance of noise in HIV-1 latency control.

## 5. HIV-1 Transcription Is Stochastic in Basal and Tat-Activated Conditions 

The experiments described above provided arguments that the HIV-1 promoter functions in a noisy manner, producing transcriptional bursts [16,98,99,100]. However, these studies used GFP or destabilized GFP as reporters, with 20 h and 2 h half-lives, respectively, and they required the viral RNA to be spliced, exported and translated. In addition, the inclusion of the Tat feedback loop in the reporters added complexity to the system and could confound interpretations. Therefore, it was important to directly observe HIV-1 transcription in live cells. This was achieved by labeling viral RNAs with the MS2 Coat Protein (MCP), which binds a specific RNA stem-loop of 19 nucleotides with sub-nanomolar affinity [101]. MCP-GFP fusion is expressed in cells together with a reporter RNA tagged with multiple copies of its cognate binding sites, called MS2. The MCP-GFP associates with these sites and enables visualization of the reporter RNA in live cells, including at its transcription site (Figure 3B). Single-molecule sensitivity is achieved by using 24 or more MS2 sites [102,103,104]. Using HeLa cells containing a single copy of the HIV-1 reporter tagged with 128 MS2 sites (Figure 3A,B), it was observed that the brightness of the HIV-1 transcription site fluctuated stochastically over time, providing direct evidence that the viral promoter is subject to noise. This was the case in both the presence and absence of Tat. A detailed view of viral transcriptional dynamics was obtained by imaging at high temporal resolution (one image per 3 s) and for long time periods (up to 10 h). Long-term imaging was facilitated by the increased brightness of the 128 MS2 RNA tag as compared with the traditional 24 MS2 tag [103], helping to decrease photobleaching and enabling single RNA molecules to be imaged for about five times longer [20].

When Tat was constitutively expressed in trans, viral transcription showed frequent bursts of activity lasting a few minutes. A burst consisted of a group of closely spaced polymerases, called polymerase convoys, which were transcribed together through the viral genome (Figure 3C,D). Convoys were separated by inactive periods of 1–2 min and contained 3 to 40 polymerases spaced by about 260 nucleotides on average, but sometimes as close as 60 nucleotides. Molecular analyses showed that convoys are formed by a re-initiation mechanism facilitated by the mediator complex. Interestingly, an analysis of the mechanical forces suggests that the spacing between polymerases in the convoys is likely to be maintained by the DNA torsional stress created by the elongating polymerases [105]. Long-term imaging revealed that the HIV-1 promoter can also occasionally enter long inactive periods lasting for more than 30 min (Figure 3D).

Importantly, studies of the same reporter in the absence of Tat showed that the viral promoter is mostly inactive but occasionally produces bursts of transcription lasting 5–10 min and forms a few successive polymerase convoys (Figure 3E) [21]. This was surprising, as in the absence of Tat every polymerase is expected to enter a paused state to limit the viral transcriptional output, thereby precluding the formation of convoys containing closely spaced polymerases (one every few seconds). These data demonstrated that the HIV-1 promoter displayed spontaneous transcriptional fluctuations, which could be responsible for the phenotypic bifurcations observed in the GFP-tagged viruses that had the entire Tat feedback loop. In agreement with previous studies [16,63,64,70,95,96], it was proposed that the rare and stochastic transcriptional bursts observed in the absence of Tat may be amplified by Tat to trigger latency exit, while, in contrast, the long and rare inactive periods observed in the presence of Tat could lead to Tat depletion and induce latency [20]. 

## 6. Stochastic Transcriptional Response to Activation

The HeLa cell line with the 128 MS2-tagged HIV-1 reporter was used to study the response of the viral promoter to TNF-α activation [23]. In the absence of Tat, individual cells showed heterogeneous responses to TNF-α stimulation in spite of a homogeneous nuclear translocation of NF-κB. Some cells did not respond at all, while others responded either synchronously with a peak of transcription after ~20 min, or later and at lower levels. Interestingly, the fast-responding cells had a high probability to reactivate transcription after a second pulse of TNF-α, maintaining a “quick responder” state for about 3 h after the first stimulation [23]. This result suggested that a fraction of the cells were in a “pre-activated” state, which was more favorable for transcriptional activation by NF-κB, and likely corresponded to a particular molecular state of the viral promoter. 

Importantly, TNF-α was also found to induce heterogeneous transcriptional responses of the HIV-1 promoter in Jurkat T cells [106]. In this case, the use of the LGIT reporter with GFP and the intact Tat feedback loop showed that the initial noisy response to TNF-α was subsequently amplified by Tat, leading to viruses being either fully activated or not responding at all [106]. This highlights the links between single-cell heterogeneity in the viral basal state, which drives a noisy TNF-α response, and noise amplification by the Tat feedback loop leading to stochastic viral activation. These observations likely explain the stochasticity of the exit from latency in patients where only a fraction of latent cells become activated by stimulating agents, with repeated rounds of stimulation activating new latent viruses every time [67].

## 7. Factors Influencing HIV-1 Transcriptional Noise

The studies above demonstrated that the HIV-1 promoter is noisy and that this noise affects latency. To better understand this noise, the factors and processes creating or affecting it were characterized using three single-cell methods: indirect live-cell imaging of HIV-1 transcriptional activity using GFP-tagged viruses (mainly LTR-GP-IRES-Tat-LTR (LGIT) reporter or Nef-GFP viruses); direct live-cell imaging of transcription using arrays of MS2 tags; and single-molecule FISH (smFISH) in fixed cells. 

### 7.1. Promoter Architecture Controls Transcription Stochasticity 

Intrinsic noise is created by a variety of stochastic events affecting transcription, including the stochastic binding and dissociation of transcription factors, nucleosomes and nucleosome-modifying enzymes (reviewed in [107,108]). These factors bind and dissociate via different dynamics, creating transcriptional fluctuations at various time scales from minutes to days [20]. The binding dynamics of transcription factors has been studied *in vivo* with a variety of techniques, such as FRAP, dynamic ChIP and single-molecule tracking (for review see [109,110,111]). These experiments indicate a long binding half-life for nucleosomes, in the order of hours; a medium half-life of minutes to hours for few key factors such as TBP; and a fast half-life of a few seconds for most transcription factors [107,109,111]. Of interest for HIV-1, recent dynamic ChIP experiments indicate that SP1 has a turn-over rate on most cellular promoters in the order of minutes, and slow/fast biphasic behavior at enhancers and other distinct from promoters sites [112]. Single-molecule tracking of NF-kB showed that this factor has a residency time on DNA in the order of 5–10 s [113].

TBP is an essential factor that binds with high affinity to TATA box sequence and nucleates the assembly of the pre-initiation complex. As mentioned above, the HIV-1 promoter contains a TATA box (Figure 1). The transcriptional activity of mutant viral promoters with decreased TBP binding was studied using direct transcription imaging with MS2-tagged reporters [20]. These mutants demonstrated an increased probability of long inactive OFF states, which were nonpermissive for transcription and could last for hours, possibly leading to a depletion of Tat that could disrupt the feedback loop and facilitate a switch to viral latency [20]. Consistently, in an elegant mutagenesis screen performed in the context of a Nef-GFP virus in Jurkat T cells, it was found that TATA box mutant viruses had a higher switching rate between high (acute) and low (latent) GFP states [96].

The Sp1 and NF-κB binding sites of the HIV-1 promoter play essential roles in basal and activated transcription ([71] and references therein). To elucidate how they influence noise and latent/acute viral switches, Jurkat cells were transduced with LTR-GFP-IRES-Tat-LTR vectors carrying single mutations of Sp1 or NF-κB binding sites, and GFP expression was followed for 21 days in a population of cells with single-copy integrations [95]. The results showed that the Sp1 sites contribute to HIV-1 phenotypic noise and influence the switches between metastable high (acute) and low (latent) GFP states. Single mutations of Sp1 binding sites led to more frequent switches between these states, with the mutation of the Sp1 site III displaying the strongest phenotype. This suggests that SP1 stabilizes both the latent and transactivated states and thus plays a dual role in activation and repression [95]. This was confirmed using a full-length Nef-GFP virus and the previously mentioned mutagenesis screen aiming at identifying fast-switching HIV-1 mutants [96]. ChIP experiments showed that the Sp1 sites differentially recruit histone deacetylase HDAC1 to latent and activated promoters, possibly explaining how it stabilizes the latent state [95]. The two NF-κB binding motifs were found to have different functions since only the mutation of NF-κB site I decreased the number of cells in the activated GFP state. These observations were consistent with ChIP results showing that in the latent but not transactivated state, NF-κB site I was essential for the recruitment of the RelA(p65)-p50 heterodimer, leading to transcriptional activation, while NF-κB site II rather recruited a repressive p50-p50 homodimer [95]. 

Interestingly, most of the HIV-1 subtypes contain two NF-κB binding sites in their promoters, with only HIV-1E having one and HIV-1C having three to four sites. Comparative studies of transcriptional noise in Jurkat cells using LGIT vectors with 0 to 4 NF-κB binding sites showed that the increase in the sites number was associated with higher transcriptional activity, while the presence of two to four sites led to attenuated noise. This might have implications for the latency regulation of different viral subtypes [114,115] 

### 7.2. Influence of Chromatin and Integration Sites on the Stochasticity of Transcription 

As already mentioned, the nucleosomal structure of the HIV-1 provirus is well defined with two nucleosomes precisely positioned 450 nucleotides before the TSS (Nuc-0) and immediately after (Nuc-1; Figure 1) [73]. The overall chromatin structure of the viral promoter is a key element of HIV-1 transcriptional regulation, with Nuc-1 remodeling by Tat playing a crucial role [73,87,97]. 

The role of chromatin in noise was first probed with a viral vector lacking Tat and encoding a destabilized version of GFP (d2GFP). This virus was integrated in 30 different genomic locations in Jurkat cells and the expression noise was measured via flow cytometry. It was found that the noise depended on the viral integration site, with the mean expression and noise levels varying independently between clones [98]. Chromatin accessibility at the different integration sites was measured using DNAse I mapping. A compact chromatin around Nuc-1 correlated with higher expression noise, whereas low-noise promoters had a more open chromatin around the TSS [100]. Chromatin is dynamic and subject to stochastic fluctuations, as seen via electron microscopy and single-molecule footprinting [116]. It was thus proposed that the chromatin density reflects the transition rates between active and inactive states of the promoter, with the more closed promoters switching less frequently to active states [100]. 

The role of chromatin was also studied in the context of GFP-tagged viruses having the full Tat feedback loop, using the LGIT reporter or full-length Nef-GFP virus and integrating them at various locations in Jurkat cells [106,117,118]. As mentioned in the paragraph 6, it was found, using live-cell imaging, that the infected clones showed different switching rates from latent to activated states upon TNF-α stimulation. Depending on the integration site, the clones activated GFP expression more or less rapidly and with either a small fraction or a majority of cells activated in each clone, leading to a classification of the clones as either “low-activating” or “high-activating”. Importantly, it was shown that these differences corresponded to the chromatin state of the promoter, with more open chromatin, as measured by the fraction of acetylated histone H3, leading to more rapid, efficient and homogenous responses [106]. Moreover, the modification of histone acetylation with HDAC inhibitors induced a switch of the clone phenotype from “low-activating” to “high-activating”, confirming that chromatin state is a strong regulator of viral reactivation, both across clones and within the same clone [106,117]. In essence, these data connect changes in expression induced by TNF-α stimulation in a given cell with the initial chromatin state of the virus in this cell [118].

### 7.3. Polymerase Pausing and Viral Transcriptional Stochasticity 

The main role of Tat is to recruit P-TEFb on nascent viral RNAs, leading to the release of the polymerases paused near the promoter. The effect of pausing on HIV-1 transcriptional noise was studied using MS2-tagged HIV-1 reporters under variable levels of Tat expression in trans [21]. As mentioned in the paragraph 5, it was found that in the absence of Tat the viral promoter generated transcriptional bursts, consisting of successive polymerase convoys, and that, surprisingly, these convoys were similar to the ones observed under high Tat levels. This is incompatible with a model in which every polymerase enters a rate-limiting pause in the absence of Tat. It rather indicates a promoter-switching mechanism, with the switch possibly being a polymerase stochastically entering a long pause and preventing new initiations by blocking incoming polymerases [21,119,120]. RNAPII ChIP following a triptolide time-course confirmed a long, sub-hour-range residency time of paused polymerases at the viral promoter in the absence of Tat [21]. These data suggest that pausing is stochastic and, as a consequence, not every polymerase that initiates transcription enters a paused state, even in the absence of Tat (Figure 4D). The long pauses probably do not correspond to maturing polymerases, but could rather be analogous to inactive promoter states. Overall, stochasticity in pausing induces transcriptional bursts in the basal state, which, combined with the Tat amplification loop, generate stochastic switches between basal (latent) and transactivated (acute) viral states. 

## 8. Modeling of HIV-1 Transcription 

The dynamic binding of nucleosomes and transcription factors to DNA creates intrinsic noise in transcriptional output [107,108]. Mathematical models describe these dynamics and provide a mechanistic understanding of the transcriptional noise. These models are used to determine the molecular states of the promoter and the transition rates between the states, through the accurate fit of the observed fluctuations in the transcriptional output. Combining the kinetic and molecular characterization of promoters allows the best model to be chosen, helping to identify the sources of noise and to understand promoter function and regulation. 

Promoters bind a multitude of factors and harbor a variety of chromatin marks and nucleosome positions. This molecular diversity largely exceeds the number of kinetic transitions that can be identified from the data, which rarely exceeds three or four. However, it is reasonable to admit that the experimentally identified transitions correspond to rate-limiting steps, which are often the key steps regulating promoter function. The study of a promoter under a variety of conditions can reveal additional limiting steps and enables the development of more complete models [6]. In this respect, the HIV-1 promoter is unique since its noise has been extensively studied using several experimental approaches and under a plethora of conditions: with and without Tat, which was provided in cis or in trans; after treatment with a variety of activating signals; in different cell lines; and integrated at many chromosomal locations. As a result, a number of models have been developed for the HIV-1 promoter and shown to be relevant in some conditions. However, no unified model has been proposed yet. 

For promoter modeling, important considerations are the experimental approaches used to measure noise and the acquisition conditions. Direct transcriptional imaging using MS2 arrays enables single-molecule sensitivity and yields extremely precise information on promoter dynamics. However, because the polymerase dwell time at the transcription site is at most a few minutes [104], this method requires a high imaging rate, leading to bleaching and phototoxicity that make imaging beyond 12 h difficult. Therefore, it is not well adapted for slow fluctuations occurring over days, as for instance a switch of viruses with the Tat feedback loop from high to low metastable expression states [63,70,95,96]. Nevertheless, robust methods have been developed to directly extract the number of promoter states and their transition rates from the MS2 data, as well as to compare many different models [121]. The smFISH approach gives accurate single-molecule information and has the advantage of producing datasets for thousands of cells. It however lacks temporal information, which needs to be inferred from the known RNA half-life in combination with modeling. This method can thus be misleading if not combined with live imaging. GFP-tagged viruses have the advantage of enabling long-term imaging, over days to even weeks, and consequently represent the most appropriate approach to characterize very slow fluctuations. Conversely, as mentioned in the paragraph 5, the long half-life of GFP (20 h to 2 h for the wild-type and its destabilized variants), buffers rapid and medium transcriptional fluctuations, which can be overlooked. GFP, in addition, only indirectly provides information on transcription and should be ideally combined with an RNA-imaging method. 

It is important to mention that the above approaches yielded different models of transcription. While a simple two-state ON/OFF (random telegraph) model (Figure 4A) was able to fit the data in the studies using GFP reporters [16,98] or RNA detection via smFISH [100], it was unable to fit the data with the MS2-tagged reporters and more complex models were required [20,21,22,23]. This difference can be explained by the fact that the MS2-MCP approach can reveal additional rate-limiting steps of transcription due to its direct detection of RNA coupled to high temporal resolution. Furthermore, different imaging timescales enable different regulatory mechanisms to be revealed [20].

## 9. Stochastic Models of HIV-1 Transcription without the Tat Feedback Loop

Models describing the HIV-1 promoter in the absence of the Tat feedback loop have been derived from direct transcription imaging using the MS2 system, in basal conditions or in the presence of Tat, which was provided in trans, as well as under TNF-α activation. As mentioned above, the data were fitted with more complex three-state models with several topologies: linear, branched or circular (Figure 4B–E) [20,21,23]. The linear and branched models (Figure 4C,D) are mathematically equivalent and can be distinguished only with additional molecular analyses. When Tat was provided in trans, the linear model was well suited and could fit the data from both wild-type and TATA-box mutant promoters (Figure 4B). In this case, the first OFF state corresponds to a promoter with bound TBP and not occupied by nucleosome, while the second deeper OFF state is a nucleosome-repressed state (Figure 3D and Figure 4B) [20]. 

In the absence of Tat, traditional models of pausing in which all polymerases undergo a pause did not fit experimental data (Figure 4C; [21]). Instead, the data fitted a branched model where the Tat-dependent pause is facultative and represents one of the model branches, the other being a nucleosome-repressed state (Figure 4D). When compared to high-Tat conditions, Tat was found to (i) decrease the probability of facultative pausing; (ii) increase the rates of pause release for the fraction of pausing polymerases; and (iii) stimulate transition from the nucleosome-repressed to the active state (Figure 4D) [21]. 

The HIV-1 promoter was also modeled under nonstationary conditions during TNF-α activation. In this case, two-state models with their rates depending on NF-kB concentrations did not fit the rapid transcriptional activation and shut-off that were observed. Instead, a directional cyclic model with a refractory state could accurately fit the data (Figure 4E) [23]. The nature of the refractory state was not identified, but it could be due to a cell-wide feedback mechanism in the NF-kB system [122], or to a refractory state of the promoter itself. 

The studies above yielded promoter transition rates in the order of minutes for the fast transitions (i.e., pausing in high-Tat conditions) and tens of minutes to hours for the slow transitions (pausing in the absence of Tat or chromatin opening in Figure 4B) [20,21]. The variety of models required to describe the HIV-1 promoter in different conditions highlights the diversity of the molecular states of the promoter and its distinct rate-limiting steps. Clearly, further efforts are required to generate a unified model that recapitulates the full range of HIV-1 promoter states and their dynamics.

## 10. Modeling the Tat Feedback Loop and Bistability versus Bimodality

The fact that the Tat feedback loop drives two alternative viral states raises the question of whether this circuit, which fuels its own activity and lacks a virally encoded repressor, is bistable [63,70]. Mechanistically, bistability requires self-cooperativity in the action of Tat, which can be caused by different mechanisms: homo-multimerization, cooperative binding, or a multi-step activation mechanism with Tat acting at several steps. According to mathematical modeling, simple positive feedback loops without cooperativity lack bistability [62,123,124,125]. These simple feedback loops can, however, show bimodality in the absence of bistability [125]. Bimodality without bistability generates populations with unstable high and low expression states or, alternatively, a single metastable state and an unstable one. Bimodality dependent on positive feedback differs from the bimodality of slowly switching promoters (for the latter see, for instance, [12]) by the lifetimes of the metastable states. In the presence of feedback, these lifetimes can be much larger and are not limited by the promoter-switching rates [65]. In the case of HIV-1, this emphasizes the role of feedback in the maintenance of latency. 

**Figure 4 viruses-15-01969-f004:**
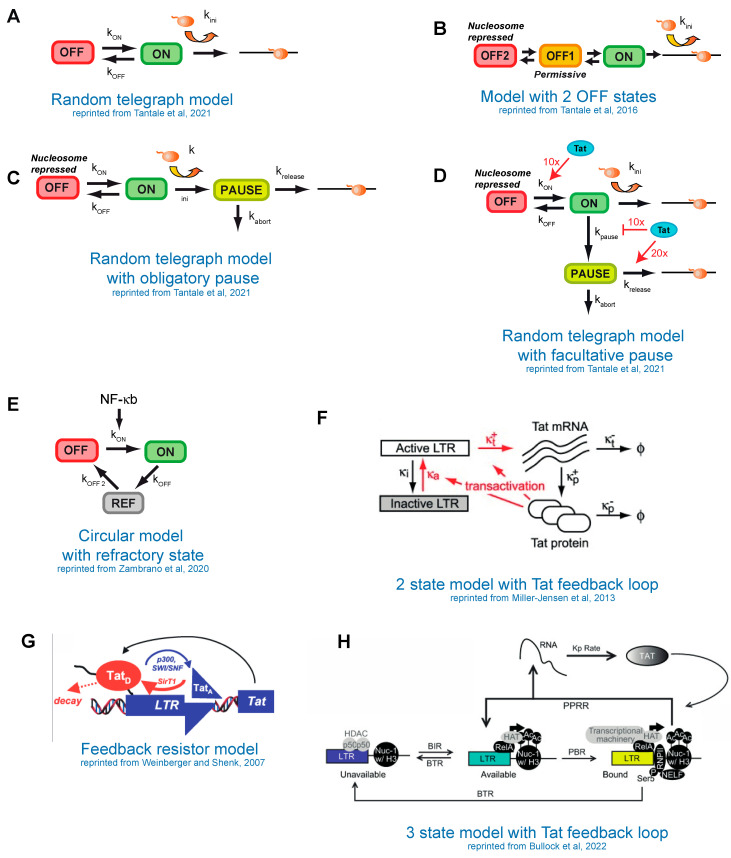
Different models of HIV-1 transcription. (**A**) Random telegraph model; (**B**–**E**) models without Tat feedback loop, Tat is expressed in trans in (**F**–**H**)—models with active Tat feedback loop. (**H**) BIR—burst initiation rate; BTR—burst termination rate; PBR—polymerase binding rate; PPRR—Promoter-Pausing Release Rate. (**A**,**C**,**D**) Reprinted from Tantale, K.; Garcia-Oliver, E.; Robert, M.-C.; L’Hostis, A.; Yang, Y.; Tsanov, N.; Topno, R.; Gostan, T.; Kozulic-Pirher, A.; Basu-Shrivastava, M.; et al. Stochastic Pausing at Latent HIV-1 Promoters Generates Transcriptional Bursting. *Nat. Commun.* **2021**, *12*, 4503, doi:10.1038/s41467-021-24462-5 [21]. (**B**) Reprinted from Tantale, K.; Mueller, F.; Kozulic-Pirher, A.; Lesne, A.; Victor, J.-M.; Robert, M.-C.; Capozi, S.; Chouaib, R.; Bäcker, V.; Mateos-Langerak, J.; et al. A Single-Molecule View of Transcription Reveals Convoys of RNA Polymerases and Multi-Scale Bursting. *Nat. Commun.* **2016**, *7*, 12248, doi:10.1038/ncomms12248 [20]. (**E**) Reprinted from Zambrano, S.; Loffreda, A.; Carelli, E.; Stefanelli, G.; Colombo, F.; Bertrand, E.; Tacchetti, C.; Agresti, A.; Bianchi, M.E.; Molina, N.; et al. First Responders Shape a Prompt and Sharp NF-κB-Mediated Transcriptional Response to TNF-α. *iScience* **2020**, *23*, 101529, doi:10.1016/j.isci.2020.101529 [23]. (**F**) Reprinted from Miller-Jensen, K.; Skupsky, R.; Shah, P.S.; Arkin, A.P.; Schaffer, D.V. Genetic Selection for Context-Dependent Stochastic Phenotypes: Sp1 and TATA Mutations Increase Phenotypic Noise in HIV-1 Gene Expression. *PLoS Comput. Biol.* **2013**, *9*, e1003135, doi:10.1371/journal.pcbi.1003135 [96]. (**G**) Reprinted from Weinberger, L.S.; Shenk, T. An HIV Feedback Resistor: Auto-Regulatory Circuit Deactivator and Noise Buffer. *PLoS Biol.* **2007**, *5*, e9, doi:10.1371/journal.pbio.0050009 [123]. (**H**) Reprinted from Bullock, M.E.; Moreno-Martinez, N.; Miller-Jensen, K. A Transcriptional Cycling Model Recapitulates Chromatin-Dependent Features of Noisy Inducible Transcription. *PLoS Comput. Biol.* **2022**, *18*, e1010152, doi:10.1371/journal.pcbi.1010152 [118]. See the text for details.

*In vitro* experiments showed that Tat acts as a monomer and single-cell measurements indicate that the Tat circuit does not exhibit cooperative effects, therefore it is likely not bistable [123]. Instead, live cell data of GFP-tagged viruses are consistent with a situation in which a single stable latent state could transiently turn on the Tat feedback loop and then relax back to the latent state. This type of bimodality can be captured by a two-state model [66] and was originally suggested to be generated by cycles of Tat acetylation and deacetylation, which would delay Tat activation and act in the circuit as a dissipator to favorize relaxation to the latent state [63,123,124]. Indeed, Tat is known to be acetylated by cellular HATs p300 and hGCN5 and this regulates its transactivation potential (Figure 1, bottom panel) [126,127,128,129,130]. Lys 28 acetylation promotes Tat binding to P-TEFb and Tat-TAR-P-TEFb assembly [126,130], while Lys 50 acetylation leads to Cyclin T1 dissociation from TAR RNA [126,127,128,129]. Tat acetylation also facilitates recruitment of the SWI/SNF chromatin-remodeling complex that displaces the inhibitory nucleosome Nuc-1 [85,90]. Tat is deacetylated by the deacetylase Sirtuin 1 (SIRT1) and this leads to its recycling (Figure 1, bottom panel) [131].

A numerical analysis using a promoter model with a single ON state showed that, even in this case, the Tat acetylation–deacetylation cycle is sufficient not only for bimodality but also to maintain the stability of the inactive loop state (Figure 4G), unlike models with cooperativity that constantly switch back and forth between active and inactive loop states [123]. The pharmacological inhibition of SIRT1, which increased the level of acetylated Tat, destabilized the inactive loop state [64,123]. Indirect effects, however, cannot be excluded in these experiments, as SIRT1 inhibitors also act on chromatin through histone acetylation and influence NF-κB activation [132,133]. 

The effect of the Tat feedback loop was also investigated using promoter models having two or three states and in which Tat increases the rates of one or more transition steps [65,106,118,134]. A random telegraph model with a Tat feedback loop stimulating both transcription initiation and the OFF to ON promoter transition rate (Figure 4F) recapitulated the switches of the Tat circuit between active and latent states that are observed in cell lines using GFP-tagged viruses [96]. Similarly, two- and three-state promoter models could describe the heterogeneity of the TNF-α response as observed by smFISH and GFP tagged viruses [106,134]. A more recent implementation of this model includes an obligatory pause whose release is stimulated by Tat, as well as an additional cyclic transition from pausing to a nucleosome-repressed OFF state (Figure 4H; [118]). This model accurately describes the noisy responses of TNF-α but it is in disagreement with the facultative pausing model derived from the MS2 data (Figure 4D), which could stem from mentioned above differences between the experimental approaches. This highlights the importance of systematically comparing the existing models in different studies. Overall, while most studies converge toward models with three promoter states, some dicrepancies remain on the importance of the Tat acetylation/deacetylation cycle, the steps that are stimulated by Tat and the topology of the model, which can be cyclic, branched or linear.

## 11. Manipulating HIV-1 Transcriptional Noise as a Strategy for a Cure

Transcriptional noise should be taken into consideration while searching for an HIV-1 cure. Two possible strategies of latent reservoir eradication are currently under investigation. The “block and lock” strategy aims to permanently silence all latent proviruses to prevent rebound upon interruption of the therapy [135], whereas the “shock and kill” strategy focuses on complete eradication of the viral reservoir. The latter strategy uses latency-reversing agents (LRA) to reactivate latent proviruses, leading to elimination of the infected cells by the immune system or via the cytopathic effects of HIV-1, while antiretroviral therapy prevents new infections [136]. For the moment, elimination of the viral reservoir in patients has not been achieved and one of the barriers is the incomplete reactivation of latent viruses due to the stochasticity of viral transcription. To circumvent this problem, screens of compounds modulating transcriptional noise have been performed [69,137]. It has been found that chromatin-modification inhibitors, including HDAC and Histone Methyl Transferases (HMT) inhibitors, as well as Bromodomain and Extra-Terminal (BET) motif inhibitors, enhance transcriptional noise. In addition, combining noise enhancers with transcriptional activators such as TNF-α or prostratin, which act through NF-κB, potentiated a reactivation of latency [69]. Interestingly, a recent genome-wide screen of LRA combinations confirmed a synergy of NF-κB and noise-enhancing compounds. In this case, an activator of the noncanonical NF-κB pathway associated with an HDAC inhibitor or a BET inhibitor, JQ1, synergistically activated viral expression [138]. On the other hand, noise suppressors effectively limit spontaneous reactivation of latent HIV-1 and have a potential for use in the “block and lock” strategy [69,137]. 

## 12. Perspectives

Previous studies led to two theories of latency exit: activation of latent memory T cells or spontaneous activation of the Tat feedback loop by the intrinsic noise of the viral promoter ([66] and references therein). The last two decades of research showed that the HIV-1 promoter is noisy and that the main sources of noise lie in the dynamic binding of core transcription factors, the chromatin status of the promoter and the facultative RNAPII pausing. The basal noise of the promoter generates heterogeneous responses to transcription activators such as TNF-α, and amplification of this noise by the Tat feedback loop leads to a random activation of latent proviruses in a fraction of cells. These conclusions demonstrate that intrinsic noise generates a stochastic response to cell activators and reconcile the two theories of latency reactivation. They are also in agreement with a recent model, which underscores dependence of viral reactivation on a random host-controlled phase [139]. Furthermore, the highlighted results explain the heterogeneous reactivation of latent viruses in cells from HIV-1 patients, which needs to be understood in depth to improve therapies aiming at clearing the virus from patients. Several outstanding questions remain. 

First, matching the kinetic promoter states to known molecular states of the HIV-1 promoter remains challenging. Indeed, the molecular identity of the various kinetic states deduced from live-cell fluctuation analyses remains incompletely characterized. Recent approaches such as single-molecule footprinting [140] allow all the molecular states of the promoter in a cell population to be quantitatively measured and should stimulate future progress in this area. In particular, they may help to determine what is the exact state of the viral promoter in TNF-α responders versus nonresponders, and how transcriptional bursting is regulated at different integration sites. This may lead to improvements in and a unification of the stochastic models of the HIV-1 promoter, with perhaps more predictive values when it comes to patients. In the long run, it may be beneficial to couple stochastic models of the HIV-1 promoter with models of the infection in patients to better predict viral rebounds.

A second issue is to describe HIV-1 transcriptional noise in the cell types representing the most important viral reservoirs, such as memory T lymphocytes and macrophages. Indeed, we know very little about HIV-1 transcription in these cells, as most studies on HIV-1 noise were performed in simpler models such as HeLa or Jurkat cell lines. 

The third, still poorly understood, question is the role of extrinsic noise, both in basal transcription and in the heterogeneity of the responses to activating agents. While it is clear that the molecular promoter status plays a role, the cellular status is likely to be equally important. T cells are indeed known to occur in different states that have different capabilities to sustain HIV-1 transcription [141,142], and such cell-wide features likely play an important role in patients. 

Finally, while it was proposed that HIV-1 splicing buffers transcriptional noise [143], we may wonder how other post-transcriptional processes, such as RNA nuclear export and translation, affect noise. Single-cell and single-molecule approaches, including imaging, biochemistry and sequencing, together with mathematical modeling, will be instrumental in answering these questions and bringing not only new discoveries in this exciting field but also therapeutic applications.

## Figures and Tables

**Figure 1 viruses-15-01969-f001:**
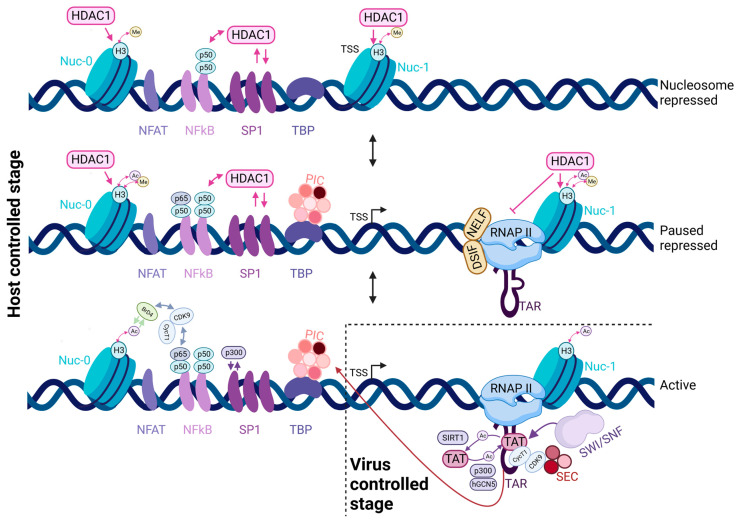
Transcriptional stochasticity of HIV-1 promoter is regulated by host factors and viral transactivator Tat. Three states of the viral promoter and two stages of transcription (host- and virus-controlled) are indicated. Nucleosome-repressed state—top; promoter-proximal pause-repressed state—middle; and activated state—bottom. During the host-controlled stage, stochastic activation of the viral promoter, influenced by the depicted factors, leads to an increase in Tat level, which triggers the Tat-controlled feedback loop to fuel viral transcription. See the text for details. Created in BioRender.

**Figure 2 viruses-15-01969-f002:**
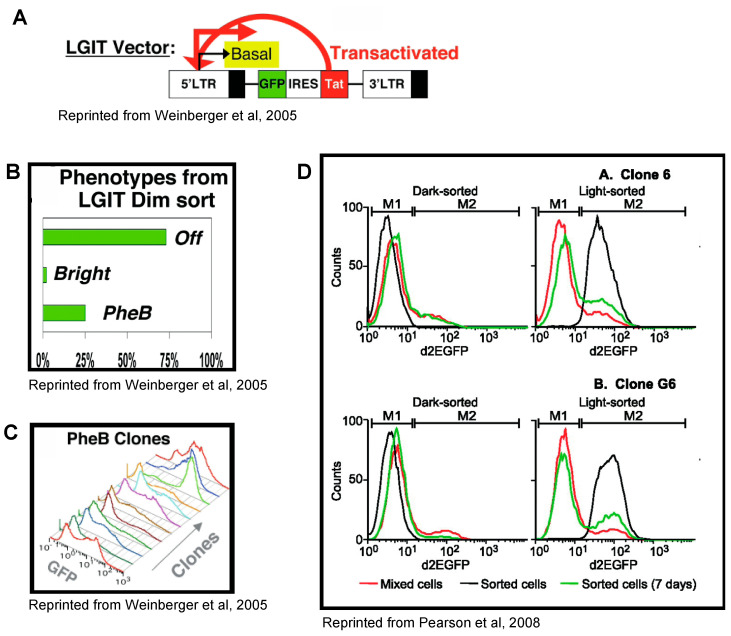
Clones with HIV-1 reporter containing Tat feedback loop display phenotypic bifurcation. (**A**) LTR GFP IRES Tat LTR (LGIT) reporter allows HIV-1 expression noise to be studied by measuring GFP level via FACS. (**B**) Jurkat clones with single integration of LGIT vector obtained via FACS sorting from low- level-GFP-expressing cells (dim sort) exhibit 3 different phenotypes after 3 weeks in culture (75% OFF; 2% Bright; 23% PheB). (**C**) Flow histograms of LGIT PheB clones. (**D**) Spontaneous reactivation and shutdown of two clones infected with HIV-1 reporter with d2EGFP in place of Nef [70]. FACS analysis of the cell populations immediately following cell sorting are shown by the black lines. The same cell populations were analyzed after 7 days (green lines). The unsorted population is shown by the red lines. (**A**–**C**) Reprinted from Weinberger, L.S.; Burnett, J.C.; Toettcher, J.E.; Arkin, A.P.; Schaffer, D.V. Stochastic Gene Expression in a Lentiviral Positive-Feedback Loop: HIV-1 Tat Fluctuations Drive Phenotypic Diversity. *Cell* **2005**, *122*, 169–182 [63], pp. 171, 173. Copyright 2023, with permission from Elsevier. (**D**) Reprinted from Pearson, R.; Kim, Y.K.; Hokello, J.; Lassen, K.; Friedman, J.; Tyagi, M.; Karn, J. Epigenetic Silencing of Human Immunodeficiency Virus (HIV) Transcription by Formation of Restrictive Chromatin Structures at the Viral Long Terminal Repeat Drives the Progressive Entry of HIV into Latency. *J. Virol.* **2008**, *82*, 12291–12303 [70], p. 12295.

**Figure 3 viruses-15-01969-f003:**
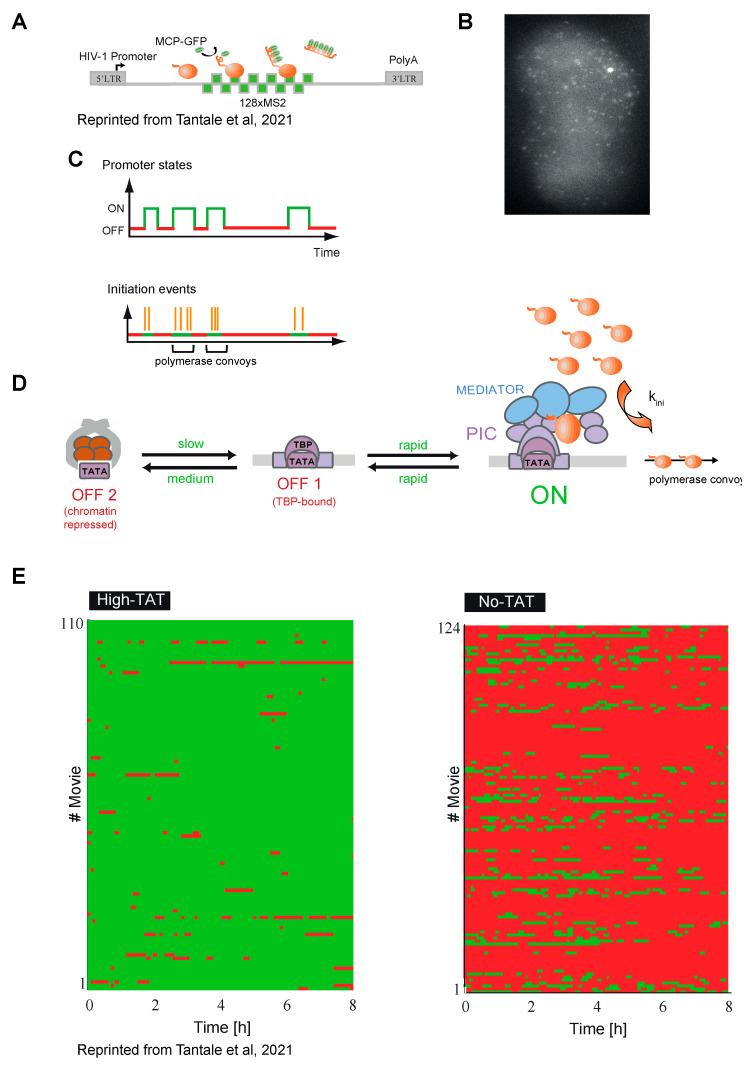
Transcriptional bursting of HIV-1. (**A**) HIV-1 reporter tagged with 128 MS2 (MCP-binding sites) allows nascent and mature HIV-1 RNAs to be detected in live cells through binding of MCP-GFP, which is co-expressed in the same cell. (**B**) Microscopic image of the cell expressing HIV-1 reporter depicted in (**A**). Bright spot is an active transcription site, small spots are single RNA molecules. (**C**) Top—promoter ON and OFF states; bottom—RNA polymerase II initiation events during the ON states, shown as orange lines, allow convoys to be formed, which are indicated. (**D**) Scheme of switching of HIV-1 promoter between chromatin-repressed OFF2 state and TBP-bound OFF1 state and ON state, during which the polymerase convoys are formed under control of the mediator complex. (**E**) Aggregated graphs show intensity of transcription site (TS) of the HIV-1 reporter during 8h long recordings under High- and No-Tat conditions, and each line represents a recording of one cell. ON states of the promoter, which correspond to one or several transcriptional bursts, are in green, and the OFF states are in red. # Movies –movie number (**A**,**E**) Reprinted from Tantale, K.; Garcia-Oliver, E.; Robert, M.-C.; L’Hostis, A.; Yang, Y.; Tsanov, N.; Topno, R.; Gostan, T.; Kozulic-Pirher, A.; Basu-Shrivastava, M.; et al. Stochastic Pausing at Latent HIV-1 Promoters Generates Transcriptional Bursting. *Nat. Commun.* **2021**, *12*, 4503, doi:10.1038/s41467-021-24462-5 [21].

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
