# Peer review of "Transcriptional Stochasticity as a Key Aspect of HIV-1 Latency"

_viruses, 2023, doi:10.3390/v15091969_

Round 1

Reviewer 1 Report

Damour et al. present a review on "Transcriptional stochasticity as a key aspect of HIV-1 latency". The authors particularly focus on the molecular aspects, discussing the different sources of stochasticity and noise and their contribution to latency during HIV-1 infection. Damour et al. introduce the HIV-1 transcriptional machinery and the factors that influence the HIV-1 transcriptional noise. The authors then provide a comprehensive overview of stochastic models with graphic schematics of HIV-1 transcription. Damour et al. conclude with a discussion on the use of noise as a strategy for a cure and perspectives.

Minor revision:
However, there are two aspects that the authors may want to explore and discuss in their review. On the one hand, there are molecular/cellular processes known that reduce noise, e.g. parallel feed-back loops in the MAPK signaling cascades (Sharifian et al., Integrative Biol. 2015) or in gene regulatory networks during development by microRNAs (Hornstein & Shormon, Nat. Genetics 2006; Herranz & Cohen, Genes & Dev. 2010) or in general (Arias & Hayward, Nat.Rev.Genet. 2006). In contrast, other studies discuss the "need for noise" by biological processes (e.g. Azpeitia et al., BMC Bioinf. 2020)

Overall, this is a nicely written review ready for publication.

Reviewer 2 Report

Overall this review by Damour et al. is a very nice, comprehensive and well-researched review article on transcriptional noise and its role in HIV latency.  A review on the role of transcriptional noise in latency is important for the field and it is also exciting to see a review on new state-of-the-art techniques that complement and ratify previous studies.  This is particularly noteworthy in this day and age when the data and findings of so many published findings are difficult to reproduce. 

The authors did a fairly good job in their attention to the prior scholarship though there are minor grammatical errors throughout that should be corrected.  Below are our specific comments which focus largely on clarification and attributions of previous findings:

1.     Lines 74-76:  requires clarification and qualification as the two-state model does indeed fit a fair portion of observed data (but not all) in mammalian cells (e.g. see citation 9), including many of the papers cited. Multi-state models appear to fit better data collected using MS2-MCP real time measurements (Citations 14-18). I advise the authors to discuss the limitations of the complementary approaches to explain the differences observed.  

2.     Line 106:  various groups have attempted to repeat the findings in citation 45 without success and careful analyses of viral evolution in patients on ART appear incompatible with ongoing replication during ART (e.g., PMID 27855060)

3.     Line 112-125:  The paragraph lays a good description of the stochastic model for HIV latency. I would suggest the authors to cite the actual papers and not only a review (i.e., cite references 71, 108, 112, 117). I would also suggest that the authors clarify that the fractional reactivation findings in patient cells (citation 48) mirrored and extended earlier findings of the phenomenon in cell culture (i.e., citation 71).

4.     Line 138: missing word: “…are required [for] RNAPII”

5.     Line 160: this statement may need qualification as the evidence is associative.  We are not aware of evidence indicating that the nucleosome in fact blocks RNAPII progression, in fact it could be the opposite (RNAPII pausing allows an otherwise variable positioned nucleosome to be precisely positioned).

6.     Line 246:  The authors may want to add citations 11 and 108 here as this mechanism was proposed in those earlier works as well.

7.     Line 289: this view of intrinsic noise driven exclusively by transcriptional initiation mechanisms was recently modified in the literature to include transcriptional elongation mechanisms (see PMID 34301855)

8.     Line 309: typo missing word “facilitate [a] viral switch…”

9.     In a number of paragraphs, earlier citations should also be acknowledged for precedent:

a.     Line 309:  cite refs 11 and 108

b.     Line 369-370: cite ref 112

c.     Line 374:  cite ref. 11

d.     Lines 445-448: ref 112 

These earlier papers and reported the same findings as in ref 15 though not with the MCP MS2 method.

10.  Lines 430-434: As in lines 74-76, this should be qualified since the two-state random telegraph model can fit smFISH and GFP data (refs. 11, 75)

11.  Line 469 & 475: for consistency remove the dash in “feed-back” à feedback

12.  Line 486:  it should be noted that the bimodality phenotype can also be captured with a topologically similar, but arguably simpler, two-state model with feedback (see ref. 117)

13.  Line 518: typo missing “for [an] HIV-1…”

see above comments 4, 8, and 13
